# Obesity-related T cell dysfunction impairs immunosurveillance and increases cancer risk

Alexander Piening [1], Emily Ebert [1], Carter Gottlieb[1], Niloufar Khojandi[1], Lindsey M. Kuehm[1], Stella G. Hoft[1], Kelly D. Pyles[2], Kyle S. McCommis [2], Richard J. DiPaolo[1], Stephen T. Ferris[1], Elise Alspach [1] & Ryan M. Teague [1]✉

Obesity is a well-established risk factor for human cancer, yet the underlying mechanisms remain elusive. Immune dysfunction is commonly associated with obesity but whether compromised immune surveillance contributes to cancer susceptibility in individuals with obesity is unclear. Here we use a mouse model of diet-induced obesity to investigate tumor-infiltrating CD8$^+$ T cell responses in lean, obese, and previously obese hosts that lost weight through either dietary restriction or treatment with semaglutide. While both strategies reduce body mass, only dietary intervention restores T cell function and improves responses to immunotherapy. In mice exposed to a chemical carcinogen, obesity-related immune dysfunction leads to higher incidence of sarcoma development. However, impaired immunoediting in the obese environment enhances tumor immunogenicity, making the malignancies highly sensitive to immunotherapy. These findings offer insight into the complex interplay between obesity, immunity and cancer, and provide explanation for the obesity paradox observed in clinical immunotherapy settings.

Obesity has reached epidemic levels in the United States and is associated with an increased risk of both solid and hematologic cancers[1,2]. According to the CDC, individuals with obesity have a higher incidence of 13 different types of cancers, and obesity is listed among the most preventable risk factors for cancer[3]. Obesity is also associated with compromised immunity, resulting in increased susceptibility to infections and dampened vaccine efficacy[4–6]. Specifically in cancer patients with obesity, tumor-infiltrating T lymphocytes (TIL) show reduced function and metabolic fitness compared to those from non-obese patients[7–9]. Because T cells are critical for detection and elimination of malignant cells prior to tumor formation[10–14], lack of vigilant immune surveillance could explain higher rates of cancer in populations with obesity. Whether obesity-related T cell dysfunction and compromised immune surveillance are linked to increased cancer risk in individuals with obesity has not been determined.

The advent of immune checkpoint blockade (ICB) immunotherapy over the past decade has revolutionized cancer treatment. Since ICB relies on the reinvigoration of a patient's own immune system, it was initially predicted that cancer patients with obesity would mount reduced responses and experience worsened survival outcomes. However, results of several clinical studies have shown that when patients are stratified by body mass index (BMI), obesity is often predictive of improved survival after treatment with ICB[15–18]. The mechanism by which patients with compromised tumor immunity may paradoxically respond better to immunotherapy (i.e., the "obesity paradox") has yet to be defined. One study proposed that obesity impairs T cell function by inducing higher PD-1 expression via a leptin-dependent mechanism, making T cells more amenable to blockade of PD-1[15]. However, subsequent studies from our lab and others did not observe elevated PD-1 in either obese mice or patients with cancer[8,19–21].

[1]Department of Molecular Microbiology and Immunology, Saint Louis University School of Medicine, St. Louis, MO, USA. [2]Department of Biochemistry and Molecular Biology, Saint Louis University School of Medicine, St. Louis, MO, USA. ✉e-mail: ryan.teague@health.slu.edu

Ultimately, there has been conflicting evidence on both sides of the obesity paradox, with studies finding obesity to have a positive, negative, or neutral influence on patient outcomes, likely depending on complex variables associated with different patient populations, cancer types, treatment regimens, and even biological sex[22]. Counterintuitively, immune dysfunction could also be a contributing factor. Under less pressure within the obese immune landscape, altered immunoediting could enhance the immunogenicity of outgrown tumors, thereby increasing their sensitivity to T cell responses boosted during ICB treatment. How obesity-related immune dysfunction might influence the process of tumor immunoediting has not been clearly defined.

For patients with obesity, weight loss could represent a potential strategy to overcome weakened immunity, and there have been exciting recent advances in the development of therapies to combat obesity. Along with diet and exercise, new obesity-fighting drugs have provided unprecedented options for weight loss. For example, semaglutide is an FDA-approved glucagon-like peptide-1 (GLP-1) analog that has been shown to induce significant weight loss in obese adults and adolescents[23,24]. However, recent findings suggest that obesity may elicit durable immune cell dysfunction that persists despite weight loss, raising questions about the efficacy of this approach as a possible therapeutic intervention[25]. Thus, the potential to restore antitumor immunity through weight loss has not been established.

In preclinical models of obesity, impaired tumor immunity has been consistently reported[8,20,21,26]. In the current study, we investigate if obesity-related T cell dysfunction can be overcome through weight loss as a strategy to improve antitumor immunity. In tumor-bearing obese mice, we found that CD8+ TIL dysfunction is not permanent and could be restored after weight loss achieved through dietary changes. In contrast, we found that equivalent weight loss achieved through treatment with semaglutide is insufficient to rescue antitumor immunity, illustrating the limited benefit of weight loss alone in the absence of improved nutrition and metabolic health. To define the broader implications of obesity-related T cell dysfunction and the possible link to cancer risk, we compare sarcoma development in lean and obese mice after carcinogen exposure. Obese mice develop tumors earlier and with greater incidence compared to lean counterparts, and this disparity is dependent on adaptive immune responses. Consistent with compromised immune surveillance and inefficient immunoediting, tumors derived from obese mice show enhanced immunogenicity and are highly sensitive to ICB immunotherapy. Thus, our study establishes obesity-related immune dysfunction as a mechanism driving cancer risk, but also a possible contributor to the obesity paradox.

## Results

### CD8+ T cell dysfunction in the obese tumor microenvironment

The current obesity epidemic is largely driven by excessive caloric intake from diets high in fat and sugar, characteristic of the "western diet." To emulate human obesity, we generated diet-induced obese (DIO) mice by feeding them a model western diet (WD) for 12 weeks. Compared to age-matched mice on normal chow (NC), those on WD had increased total body mass that was attributable to increased fat mass, quantified by whole-body NMR imaging (Fig. 1A). These DIO mice experienced additional obesity-associated comorbidities including lipid accumulation in the liver (Fig. 1B), increased liver enzymes ALT and AST, used as a readout for liver damage, and high serum cholesterol (Fig. 1C), all consistent with metabolic syndrome[27,28]. Analysis of fasting serum glucose and insulin showed similar levels in lean and obese mice, thus obese mice on WD did not exhibit hyperglycemia or hyperinsulinemia (Suppl. Fig. S1). We previously reported that obese mice fail to mount an effective antitumor T cell response, evidenced by impaired function among CD8+ tumor-infiltrating lymphocytes (TIL) in melanoma tumors[26]. To gain insight into the biological differences between endogenous CD8+ TIL from lean and obese mice, CD8+ T cells

were sorted from 14 day established B16 melanoma tumors and single-cell RNA sequencing (scRNAseq) was performed, revealing 6 distinct clusters of CD8+ TIL (Fig. 1D). Among these, clusters 1 and 3 displayed a transcriptional profile consistent with cytolytic effector function that included high expression of *Ifng*, *Prf1*, and *Gzmb* (Fig. 1E). Examination of clusters 1 and 3 revealed disparities in the expression of these effector genes, which was reduced in CD8+ TIL from obese mice (Fig. 1F). It has been suggested that obesity induces CD8+ T cell exhaustion through expression of PD-1[15]. However, genes typically associated with CD8+ T cell exhaustion like *Pdcd1* (PD-1), *Lag3*, and *Havcr2* (Tim3) were either equivalently expressed or even reduced in CD8+ TIL from obese mice on WD compared to lean mice on NC diet (Fig. 1F), and this was recapitulated at the cellular protein level (Fig. 1G). Thus, functional defects in CD8+ TIL from obese mice appear to be distinct from classically defined T cell exhaustion[29].

Unbiased gene ontology pathway analysis of CD8+ TIL revealed distinct metabolic differences between lean and obese mice (Supplementary Fig. S2). Notably, TIL from obese mice were enriched for genes involved in adipogenesis, cholesterol homeostasis, and oxidative phosphorylation, while TIL from lean mice expressed genes involved in glycolysis, IFNγ response, and allograft rejection. These data closely align with the prior discovery that the metabolic switch to glycolysis is required for T cells to acquire effector function[30]. Similar metabolic signatures were reported in CD8+ TIL from lean and obese mice, leading to the conclusion that metabolic differences influenced by the obese tumor microenvironment suppress antitumor immunity[21]. Whether this metabolic profile is permanently imprinted on CD8+ TIL is unknown, and the obesity-related mechanisms that underly these metabolic differences have not been identified.

### Diet-induced weight loss rescues CD8+ TIL effector function

To determine if CD8+ TIL function and antitumor immunity could be restored through dietary intervention, obese mice that had been on WD for 12 weeks were switched to normal chow for an additional 12 weeks (Fig. 2A). Compared to mice that were maintained on WD, those switched to NC lost weight rapidly and their body mass became equivalent to lean control mice within 2–4 weeks (Fig. 2B). This loss in body mass coincided with a normalization of serum cholesterol levels and overall liver health (Fig. 2C, D). When challenged with B16 melanoma and treated with anti-PD-1/anti-CTLA-4 immune checkpoint blockade (ICB), lean mice on NC resisted tumor progression whereas obese mice on WD showed no therapeutic response. In contrast, diet-induced weight loss restored responsiveness to immunotherapy (Fig. 2E) and rescued CD8+ TIL effector function (Supplementary Fig. S3), including CD8+ TIL specific for the tumor-associated antigen tyrosinase-related protein 2 (TRP-2) (Fig. 2F). TRP-2-specific CD8+ TIL from obese mice appeared uniquely dysfunctional and had lower expression of interferon gamma (IFNγ), granzyme B and perforin (Fig. 2F, G). By comparison, CD8+ TIL from previously obese mice that had lost weight showed restored expression of these effector molecules, particularly after ICB treatment, and appeared functionally similar to TIL from healthy lean control mice. Of note, neither lean nor obese mice demonstrated evidence of cachexia (tumor-induced weight loss) (Supplementary Fig. S4), which is associated with impaired responses to immunotherapy in patients with cancer[31]. These data suggest that dietary intervention and subsequent weight loss can reverse the metabolic syndrome, immune dysfunction, and impaired response to ICB associated with established obesity.

### Semaglutide-induced weight loss fails to improve tumor immunity

The studies above provide a proof-of-principle that diet and weight loss can improve metabolic health and tumor immunity, but they were

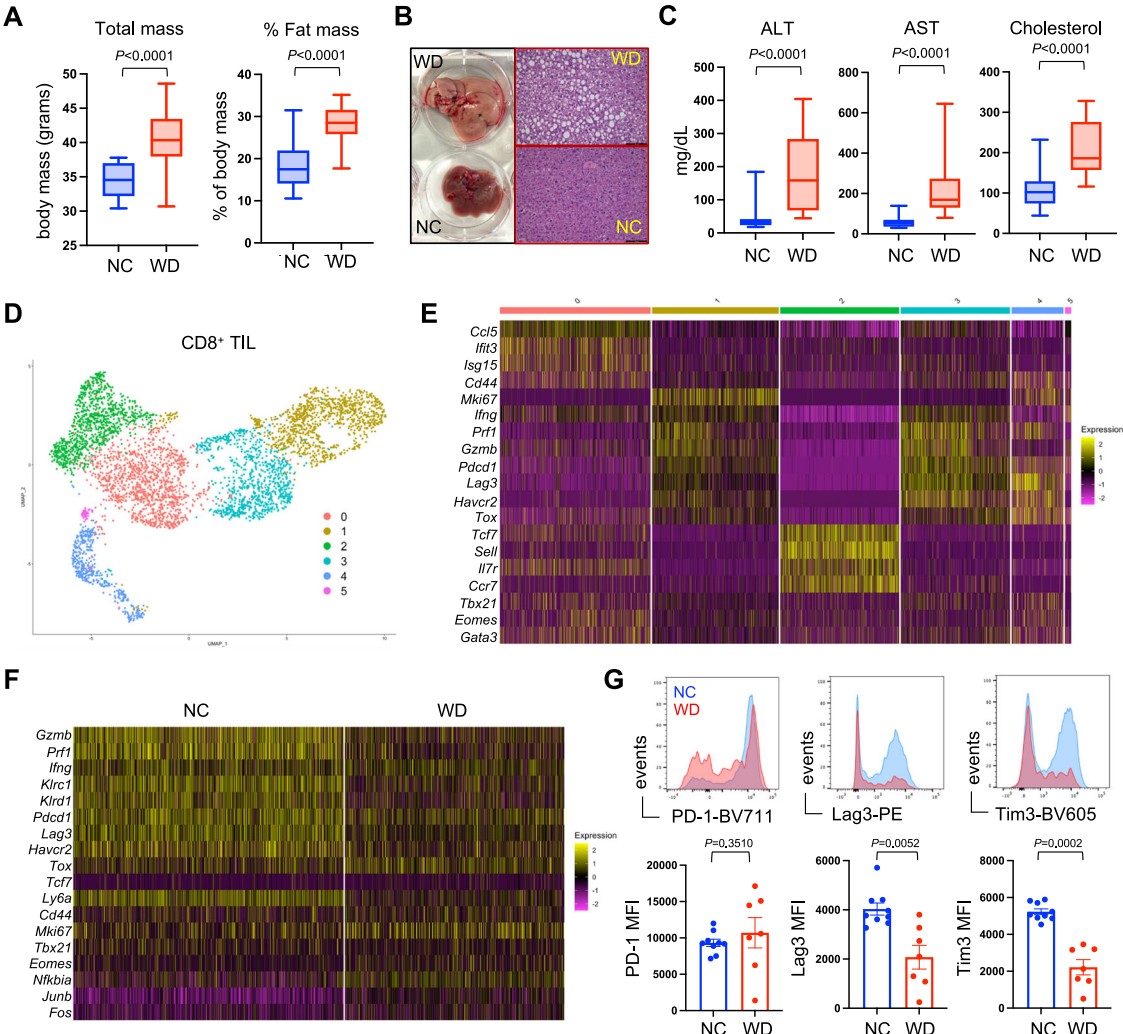

**Fig. 1 | CD8+ T cell dysfunction in the obese tumor microenvironment.** Mice were placed on either normal chow (NC) or western diet (WD) for 12 weeks. **A** Body mass (NC $n = 20$, WD $n = 20$) and fat mass (NC $n = 30$, WD $n = 30$), **B** liver histology, **C** serum liver enzymes alanine aminotransferase (ALT) and aspartate aminotransferase (AST), and cholesterol were measured (NC $n = 10$, WD $n = 20$). **D** Unsupervised UMAP clustering of scRNAseq data from CD8+ TIL sorted from day 14 established B16 tumors. **E** Relative expression of cluster-defining genes for each of the 6 clusters were identified. **F** Differential gene expression within TIL clusters 1

and 3 from mice on NC versus WD. **G** FACS analysis of CD8+ TIL from mice on NC (blue) or WD (red) showing surface protein expression of PD-1, Lag3, and Tim3 (NC $n = 9$, WD $n = 7$). Graphs display pooled data from 2 independent experiments. For all bar graphs, each point represents an individual mouse with SEM indicated by the error bars. All box-and-whisker plots: The box indicates the 25th and 75th percentile, the line indicates the data median, and the whiskers indicate the minimum and maximum of all individual values. All $n$'s represent an individual mouse. Exact $P$ values were calculated by two-sided Mann–Whitney $U$ test.

not designed to identify the source of T cell dysfunction in obese mice. The possible contributions from variables such as increased body mass, high adiposity, dyslipidemia, and altered metabolism remain undefined. Therefore, we pursued strategies to separate the variables of body mass and diet by employing the glucagon-like-peptide-1 (GLP-1) receptor agonist semaglutide; an FDA-approved drug historically used for the treatment of type-2 diabetes but now shown to elicit weight loss in people with obesity[23,24]. Obese mice were generated as before by feeding WD for 12 weeks. At that time, these mice remained on WD, but one cohort began treatment with semaglutide (WD+Sema) while the other received PBS vehicle control (Fig. 3A). Mice treated with semaglutide lost weight rapidly, achieved a body mass similar to lean mice on NC within about a week, and continued to lose weight despite being maintained on the WD throughout the study (Fig. 3B). This corresponded with lower body fat and higher lean mass that was equivalent to mice on NC (Fig. 3C). However, semaglutide-treated mice still exhibited obesity-related metabolic syndrome that included fatty liver disease and hyperlipidemia (Fig. 3D, E). Following challenge with B16 melanoma, mice that lost weight after semaglutide treatment

showed only a modest response to immunotherapy (Fig. 3F, G) and no evidence of improved CD8+ TIL function (Fig. 3H, I), which is in stark contrast to the rescue of tumor immunity observed in mice with diet-induced weight loss (Fig. 2). These results suggest that obesity-associated immune dysfunction is independent of body mass and adiposity and may instead be related to other obesity-related comorbidities.

We previously reported on the role of hyperlipidemia and lipid metabolism on immune dysfunction in patients with cancer[19], and suspected that high serum cholesterol in mice on WD (regardless of semaglutide treatment) might contribute to functional defects in CD8+ TIL. However, we found no correlation between the concentration of serum cholesterol and either TIL effector function or melanoma tumor progression in mice on WD (Supplementary Fig. S5). Additionally, inclusion of the cholesterol lowering drug atorvastatin in the WD formula failed to improve CD8+ TIL function in B16 tumors compared to those from mice WD with high cholesterol (Supplementary Fig. S5). These data indicate that high cholesterol alone does not explain the TIL dysfunction observed in obese mice.

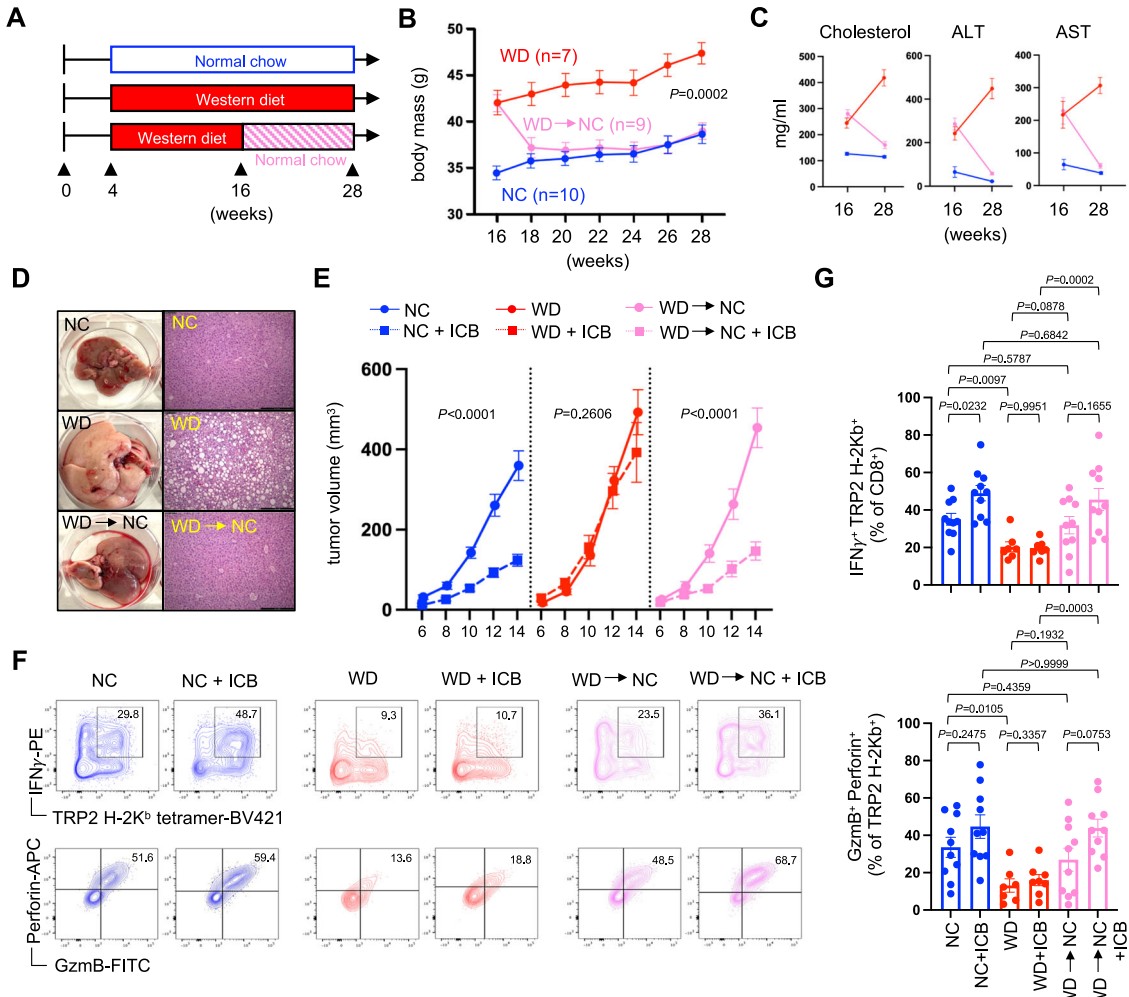

**Fig. 2 | Diet-induced weight loss rescues CD8+ TIL effector function.** Mice were placed on either normal chow (NC) or western diet (WD) for 12 weeks. **A** At that time, a cohort of mice on WD was switched to NC (WD→NC) and all mice were maintained for an additional 12 weeks. **B** Body mass was tracked over this time and graphed data is from a representative experiment showing the average mass of all mice in each group with SEM indicated by the error bars. **C** Serum cholesterol, alanine aminotransferase (ALT), and aspartate aminotransferase (AST) levels were assessed at week 16 (NC *n* = 6, WD *n* = 12, WD→NC *n* = 12), prior to diet switching, and at week 28 (all groups *n* = 10). **D** Liver histology was assessed at week 28. At week 28, mice received a subcutaneous injection of B16 tumor cells in the flank (day 0) and half of each cohort was treated with immune checkpoint blockade (ICB; anti-PD-1/anti-CTLA-4) or PBS vehicle control on days 6 and 10. **E** Tumor volumes were measured every other day starting at day 6 (NC *n* = 20, NC + ICB *n* = 20, WD *n* = 18, WD + ICB *n* = 20, WD→NC *n* = 20, WD→NC + ICB *n* = 20). **F** Tumors were harvested at day 15 and representative FACS plots show intracellular IFNγ, GzmB and Perforin expression by TRP-2-specific CD8+ TIL. **G** Graphs display pooled data from 2 independent experiments showing the percent of CD8+ TIL that are specific for TRP-2 and produce IFNγ (upper), and the percent of TRP-2-specific CD8+ TIL co-expressing GzmB and Perforin (lower) (NC *n* = 10, NC + ICB *n* = 10, WD *n* = 7, WD + ICB *n* = 8, WD→NC *n* = 10, WD→NC + ICB *n* = 10). For all bar graphs, each point represents an individual mouse with SEM indicated by the error bars. Exact *P* values were calculated by two-sided Mann–Whitney *U* test.

## Tumor immunoediting is compromised in obese hosts

The epidemiological evidence linking human obesity with increased cancer incidence is robust, but the obesity-related mechanisms that drive this elevated risk of cancer in people have not been identified[1,2]. While other studies have focused on the roles of inflammation, metabolism, and endocrine dysfunction in populations with obesity, the impact of immune dysfunction remains unknown[32]. Increased cancer risk in immunocompromised individuals and animal models have provided clear evidence that immune surveillance is critical for detecting and eliminating potentially malignant cells throughout the body and for opposing tumor progression[33–38]. As a tumor develops, T cells shape the immunogenicity of outgrown tumors by culling transformed cells that are more easily recognized and killed; a process known as immunoediting[10,13,39,40]. Our data from obese mice suggest that impaired T cell function may compromise the processes of immune surveillance and editing. Indeed, B16 melanomas from lean and obese mice display evidence of differential editing within

these distinct immune landscapes. Specifically, at the time of transplant, B16 tumor cells from in vitro tissue culture uniformly expressed the melanocyte protein Gp100/Pmel (Fig. 4A). This intracellular protein is capable of being processed and presented as a tumor-associated peptide-antigen in the context of MHC-I, or potentially MHC-II[41]. However, under these conditions, B16 tumor cells had only low levels of MHC-I and MHC-II, but expression could be induced with recombinant IFNγ stimulation (Fig. 4A). After 14 days in vivo, the phenotype of these tumor cells was disproportionally altered in lean and obese hosts. Whereas the majority of tumor cells from lean mice were low for Gp100 expression, melanoma cells in obese mice maintained high expression of Gp100 (Fig. 4B; Supplementary Fig. S6). One possible explanation for this difference is that Gp100-expressing tumor cells were more efficiently eliminated in lean recipients but not in obese mice with impaired CD8+ TIL effector function. Indeed, in obese mice where IFNγ expression by CD8+ TIL is compromised (Fig. 2), MHC-I and MHC-II expression remained low on

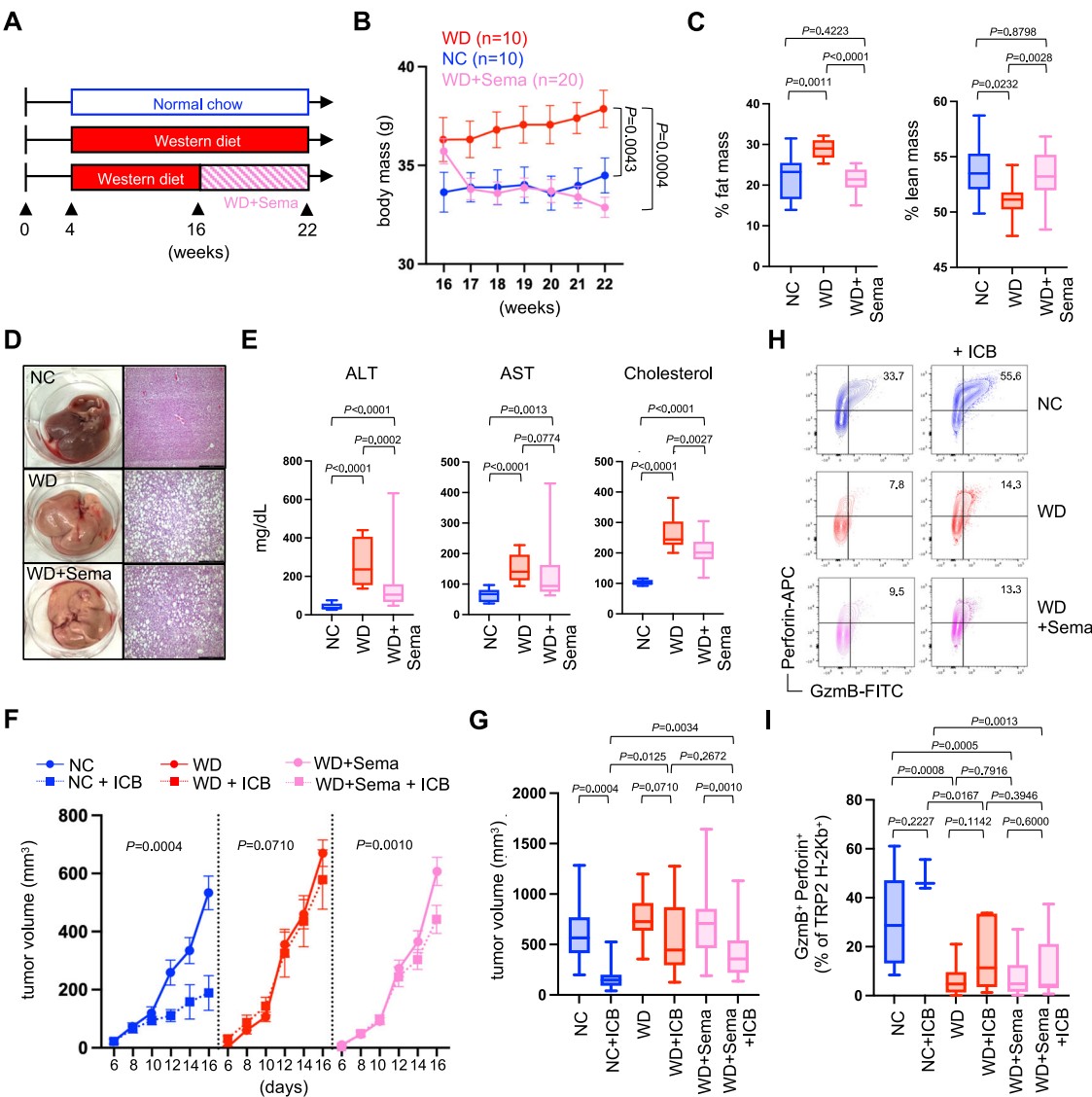

**Fig. 3 | Semaglutide-induced weight loss fails to improve tumor immunity.** Mice were placed on either normal chow (NC) or western diet (WD) for 12 weeks. **A** At 12 weeks, a cohort of mice on WD began twice weekly treatment with semaglutide (WD+Sema) for 6 weeks while being maintained on WD. **B** Body mass was tracked over time, and graphed data is from a representative experiment showing the average mass of all mice in each group with SEM indicated by the error bars. **C** After 6 weeks, fat mass and lean mass were assessed via whole-body NMR imaging (NC n = 10, WD n = 10, WD+Sema n = 20), **D** liver histology was performed, **E** and serum alanine aminotransferase (ALT), aspartate aminotransferase (AST), and cholesterol levels were measured (NC n = 10, WD n = 10, WD+Sema n = 30). At this time, mice received a subcutaneous injection of B16 tumor cells in the flank (day 0) and half of each cohort was treated with immune checkpoint blockade (ICB; anti-PD-1/anti-CTLA-4) or received PBS vehicle control on days 6 and 10. **F** Tumor volumes were measured every other day starting at day 6 and **G** pooled tumor volumes from 2 independent experiments at day 16 were compared (NC n = 18, NC + ICB n = 7, WD n = 19, WD + ICB n = 14, WD+Sema n = 35, WD+Sema+ICB n = 35). **H** Tumors were harvested at day 16 and representative FACS plots show intracellular expression of GzmB and Perforin by TRP-2-specific CD8$^+$ TIL. **I** Pooled data from 2 independent experiments show the percent of TRP-2-specific CD8$^+$ TIL co-expressing GzmB and Perforin (NC n = 9, NC + ICB n = 3, WD n = 9, WD + ICB n = 7, WD+Sema n = 17, WD +Sema+ICB n = 19). All box-and-whisker plots: The box indicates the 25$^{th}$ and 75$^{th}$ percentile, the line indicates the data median, and the whiskers indicate the minimum and maximum of all individual values. All n's represent an individual mouse. Exact P values were calculated by two-sided Mann–Whitney U test.

Gp100$^+$ melanoma cells but elevated in lean mice, and this was not the result of differences in IFNγ-receptor expression (Fig. 4C). To determine if changes in Gp100, MHC-I, and MHC-II expression could be attributed to T cell responses, we compared B16 tumors grown in normal B6 mice to those in transgenic OT-II recipients that have a severely limited T cell repertoire and should not recognize melanoma antigens (Fig. 4D). Here, the reduction of Gp100$^+$ melanoma cells and the induced expression of MHC-I and MHC-II was only observed in B6 mice with an intact polyclonal T cell compartment, and similar results were seen in a second Gp100$^+$ melanoma line (YummG) (Fig. 4E).

We interpret these results as evidence that impaired IFNγ production by dysfunctional TIL limited the expression of MHC-I and MHC-II on B16 melanoma tumor cells, hindering efficient clearance of melanoma cells in obese mice. This was partially supported in tumor-bearing lean mice treated with IFNγ neutralizing antibody, as Gp100$^+$ melanoma cells from these mice had lower surface expression of MHC-I and MHC-II but unexpectedly showed no difference in the frequencies of Gp100$^+$ cells compared to control mice (Supplementary Fig. S7). This could reflect a limitation in the experimental approach, where in vivo neutralization of IFNγ is sufficient to have a measurable effect on MHC-I expression but not tumor cell deletion.

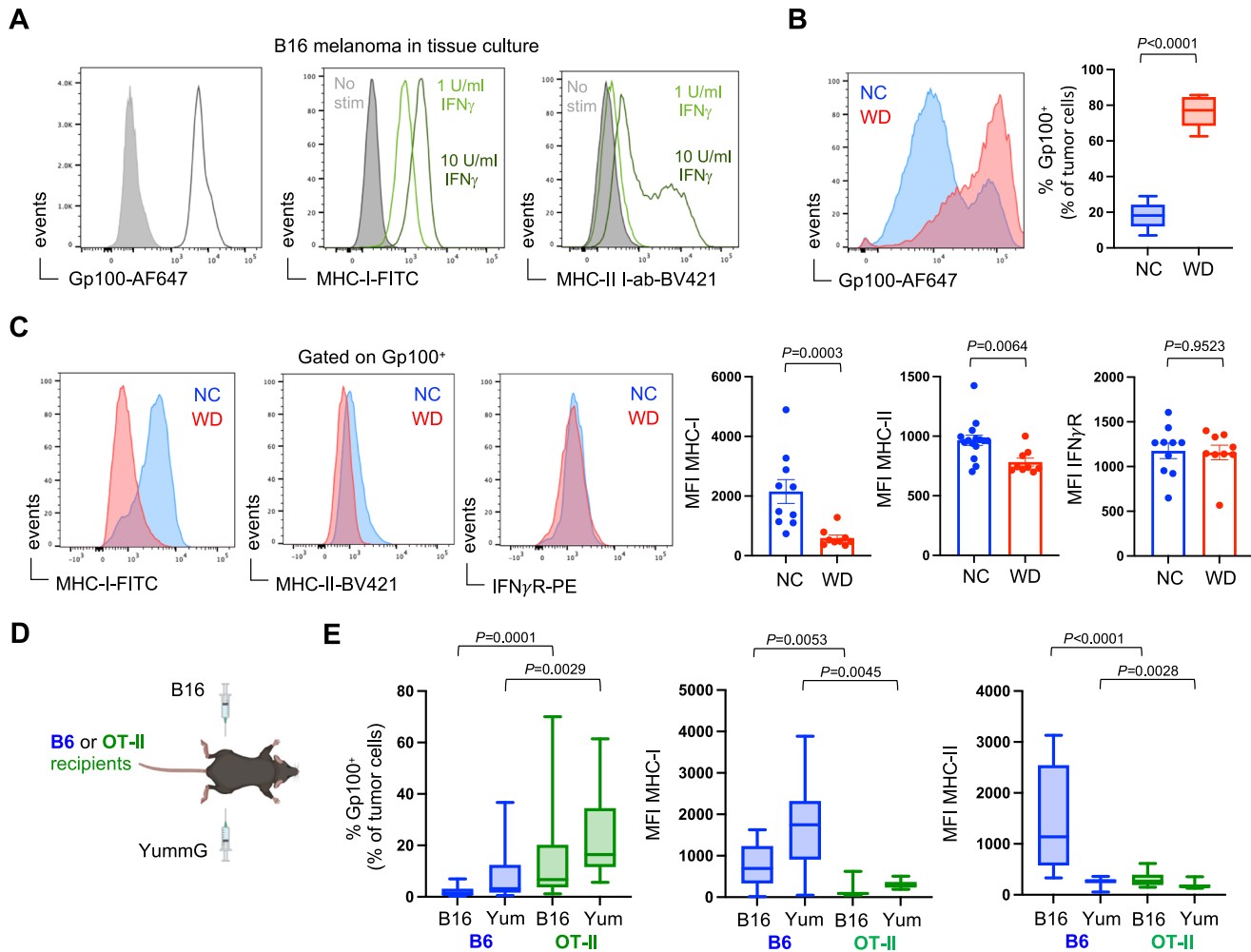

**Fig. 4 | Altered tumor immunoediting in obese hosts.** B16 melanoma tumor cells were grown and harvested from cell culture. **A** Baseline expression of intracellular Gp100, surface MHC-I, and surface MHC-II was assessed via flow cytometry. MHC-I and MHC-II expression was also analyzed after 24-h in vitro stimulation with recombinant murine IFNγ. Subcutaneous B16 tumors were harvested on day 14 and tumor cells were assessed for expression of intracellular Gp100 by flow cytometry. **B** Representative FACS plots show Gp100 expression by B16 cells from mice on normal chow (NC; blue) and western diet (WD; red), and pooled median fluorescence intensity (MFI) data from 2 independent experiments is graphed (NC $n = 15$, WD $n = 9$). **C** Representative FACS plot and pooled MFI data from 2 independent experiments is graphed showing MHC-I (NC $n = 10$, WD $n = 9$), MHC-II (NC $n = 15$, WD $n = 9$), and IFNγ receptor (IFNγR) (NC $n = 10$, WD $n = 9$) expression on Gp100+ tumor cells. **D** Bi-lateral subcutaneous B16 and YummG melanoma tumors were established in lean B6 and OT-II mice. Tumors were harvested on day 14 and expression of Gp100 and MHC-I/II was measured by flow cytometry (All groups $n = 12$ mice). **E** Graphs display pooled data from 2 independent experiments. For all bar graphs, each point represents an individual mouse with SEM indicated by the error bars. All box-and-whisker plots: The box indicates the 25th and 75th percentile, the line indicates the data median, and the whiskers indicate the minimum and maximum of all individual values. All $n$'s represent an individual mouse. Exact $P$ values were calculated by two-sided Mann−Whitney $U$ test.

Another possible source of disparate IFNγ production could stem from tumor-infiltrating natural killer (NK) cells. However, the frequency and effector function of NK cells was similar in tumors from lean and obese mice (Supplementary Fig. S8). Our data suggest that obesity-associated CD8+ TIL dysfunction and decreased production of IFNγ limits tumor MHC-I expression. An alternative interpretation is that CD8+ TIL dysfunction arises in obese mice because tumors lack MHC-I expression and are therefore poorly immunogenic, but this is not supported by expression of genes associated with activation and proliferation (*Cd44* and *Mki67*) (Fig. 1F). Furthermore, the frequencies of T cells infiltrating B16 tumors was similar in lean and obese mice, and these TIL expanded equivalently after ICB treatment (Supplementary Fig. S3). This supports our interpretation that TIL dysfunction contributes to tumor immunogenicity, with potential implications for tumor immunoediting and the response to immunotherapy.

## Immune dysfunction in obese hosts increases cancer risk

The inability to efficiently eliminate Gp100+ melanoma cells suggests that compromised immunity in obese hosts alters immune surveillance, thereby rendering them more vulnerable to developing tumors. To determine if obesity-related T cell dysfunction increases the risk of developing cancer, lean and obese B6 mice were challenged with the chemical carcinogen 3'-methylcholanthrene (MCA) and tumor incidence was tracked for four months (Fig. 5A). Unlike transplantable tumor models like B16, MCA induces de novo sarcomas over a longer time course, allowing the immune system to interact with a developing tumor from the earliest stages of oncogenesis. Here, 47% of lean B6 mice remained tumor-free by 120 days, whereas only 16% of obese B6 mice were tumor-free (Fig. 5B). Of note, tumor incidence among these obese B6 mice was similar to Rag2$^{-/-}$ mice that lack an adaptive immune system entirely (Fig. 5B, C). Tumor incidence was equivalent among all Rag2$^{-/-}$ mice regardless of diet or body mass, implying that

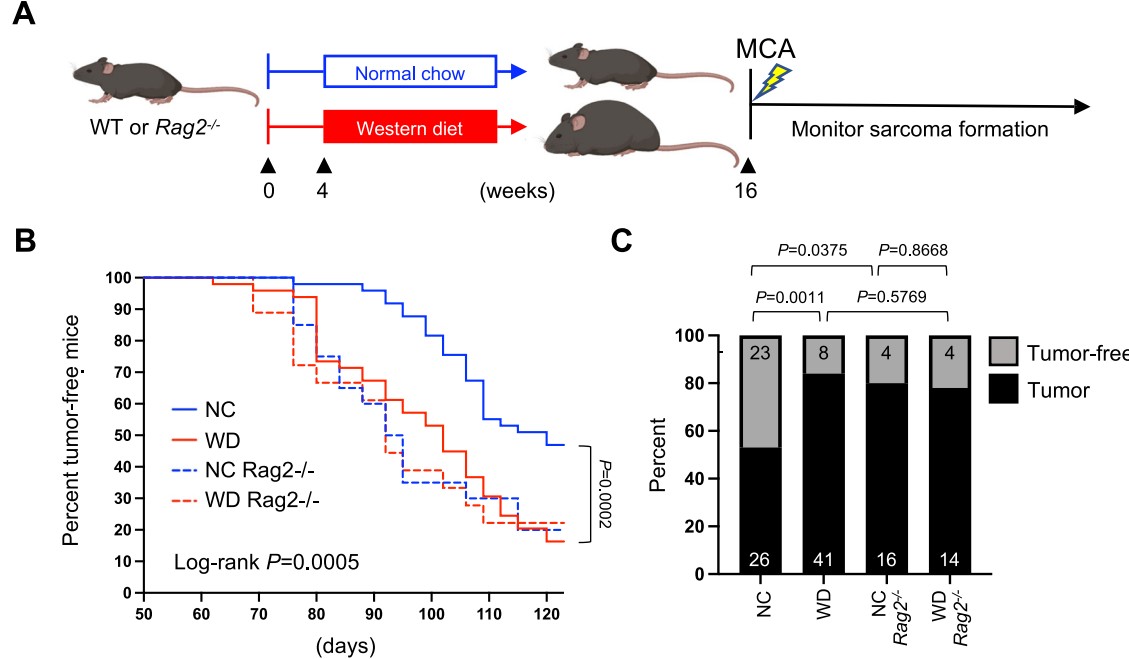

**Fig. 5 | Immune dysfunction in obese hosts increases cancer risk.** After 12 weeks on normal chow (NC) or western diet (WD), **A** wild-type (WT) B6 (*n* = 100) and Rag2⁻ᐟ⁻ (*n* = 40) mice received a single subcutaneous injection of 3'-methylcholanthrene (MCA; 50 µg) and were monitored for sarcoma tumor development out to 120 days. **B** Tumor incidence was defined as the development of a palpable sarcoma with a diameter of 8 mm and was graphed for each cohort versus time and displayed in Kaplan–Meier curves. **C** The percent and number (inset) of tumor-bearing and tumor-free mice from each cohort at 120 days is shown, with exact *P* values indicated for the bracketed groups calculated by two-sided Chi-squared analysis. **A** Created using BioRender.com.

in the absence of adaptive immunity, other obesity-related comorbidities had little influence on cancer risk. These results provide evidence that immune dysfunction in obese animals is a major risk factor for tumor development, suggesting that compromised immune surveillance could contribute to higher cancer incidence among individuals with obesity.

**Compromised immunoediting increases tumor immunogenicity in obese hosts**

Tumors that develop under different immune pressures are expected to be differentially edited, with potential impacts on tumor immunogenicity and responsiveness to therapy. To determine if tumors derived from lean and obese mice were differentially edited, sarcoma lines were generated from MCA-challenged mice and their relative immunogenicity was tested following subcutaneous transplantation into lean secondary recipients (Fig. 6A). Tumor lines derived from lean mice on NC progressed rapidly in these secondary hosts (Fig. 6B, Supplementary Fig. S9), with 18 out of 20 mice developing advanced tumors above 500 mm³ within just 30 days (Fig. 6C). In contrast, tumors derived from obese mice on WD displayed substantially slower outgrowth or were rejected in secondary hosts, with only 11 of 20 developing advanced tumors even out to day 40 (Fig. 6B, C, Supplementary Fig. S9). The disparate ability to control transferred sarcomas derived from lean and obese hosts was completely abrogated in T cell-depleted secondary recipients (Fig. 6D), confirming that differences in tumor growth were dependent on T cell-mediated immunity. Moreover, tumors derived from obese mice elicited a more robust CD8⁺ T cell immune response based on higher frequencies of Tbet⁺ Ki67⁺ and GzmB⁺ Perforin⁺ CD8⁺ TIL compared to those elicited by tumors derived from lean mice (Supplementary Fig. S10). These results suggest that tumors developing in obese hosts are more immunogenic, and thus should also be more sensitive to ICB immunotherapy. Indeed, tumors derived from obese mice were rejected in 15 of 20 secondary recipients treated with anti-PD-1, whereas tumors derived from lean mice were relatively insensitive to anti-PD-1, with only 5 of 20 mice rejecting their tumor

(Fig. 6B, C, Supplementary Fig. S9). These responses were also reflected in recipient survival after immunotherapy, which was significantly longer for recipients of tumors that had developed in the obese immune landscape (Fig. 6E). These findings demonstrate that tumors arising in obese hosts have increased immunogenicity compared to those arising in lean hosts and suggest that this may be due to decreased immunoediting by the dysfunctional adaptive immune system in obese mice. Furthermore, these results demonstrate that the differential immunogenicity of tumors edited in either lean or obese hosts is sufficient to influence the success of ICB immunotherapy.

## Discussion

Human obesity correlates with increased cancer incidence but the mechanisms underlying this relationship have not been clearly defined. Impaired immunity is a known comorbidity associated with obesity that extends to T cell dysfunction in the tumor microenvironment[7,8,20,26,42]. We employed a mouse model of DIO to define obesity-associated immune dysfunction and its impact on cancer risk, disease progression, and response to treatment. Tumor infiltration by CD8⁺ T cells was equivalent in lean and obese mice with B16 melanoma, but CD8⁺ TIL in obese mice displayed lower effector activity and this was associated with poor tumor control even after ICB immunotherapy. This immune dysfunction may be explained by the failure to transition to glycolysis, which is required for expression of effector molecules; IFNγ in particular[30]. Instead, CD8⁺ TIL in the obese tumor microenvironment maintained a metabolic signature consistent with oxidative phosphorylation, in agreement with the recent description of functionally impaired TIL in a distinct colon carcinoma model in obese mice on high-fat diet[21]. Although these data provide much needed consensus, questions still remain regarding the signals that dictate TIL metabolism under obese conditions, how TIL dysfunction influences immune surveillance and tumor immunoediting in obese hosts, and whether these TIL are permanently hyporesponsive or can be functionally rescued by reversing obesity and underlying comorbidities.

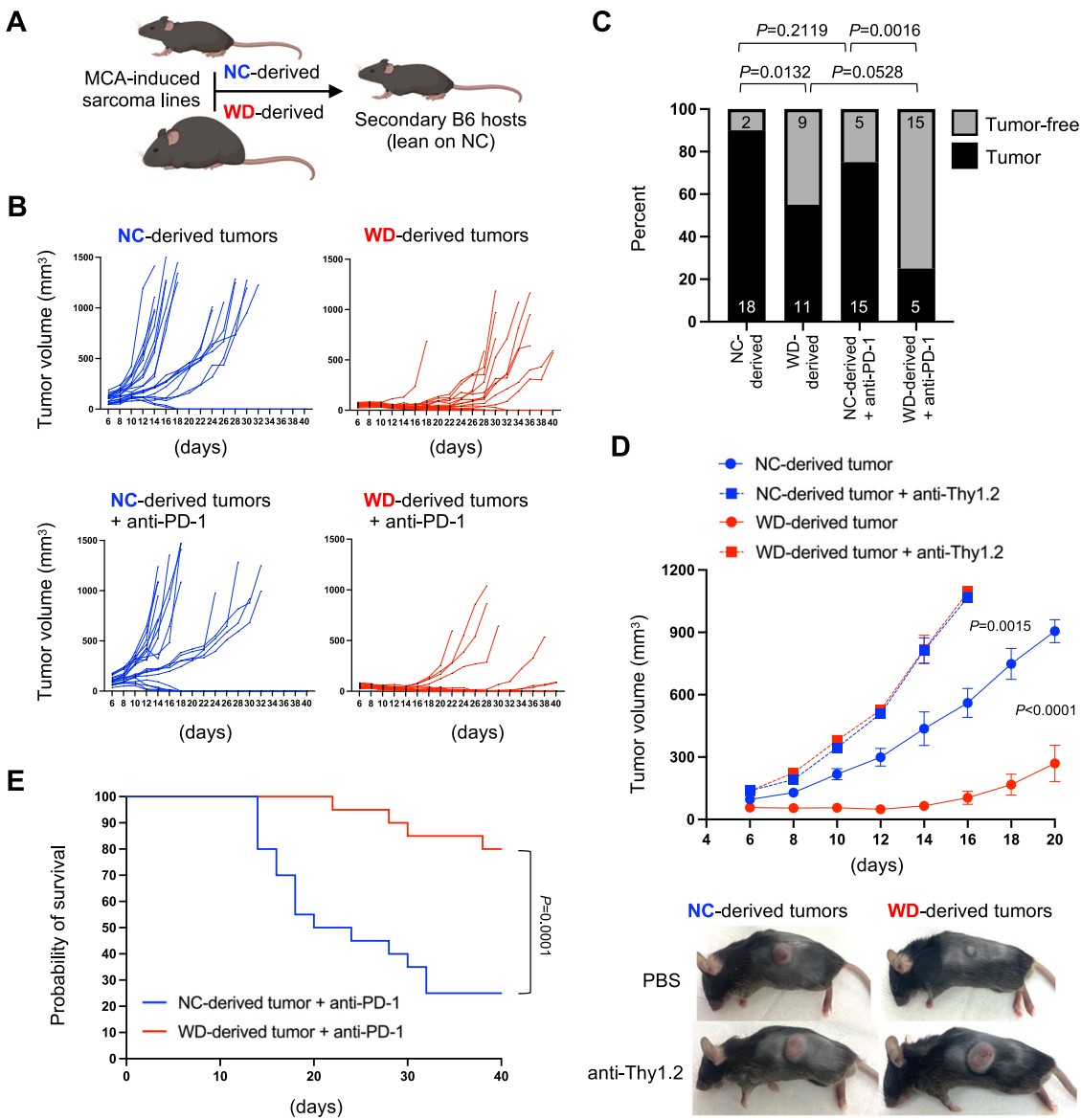

**Fig. 6 | Altered tumor immunogenicity in obese hosts.** MCA-induced tumors were harvested from wildtype mice in Fig. 5. **A** Sarcoma cell lines were generated from 4 mice on normal chow (NC) and 4 mice on western diet (WD) for reimplantation into secondary lean recipients. **B** Each NC-derived and WD-derived sarcoma cell line was injected subcutaneously into the flank of 10 secondary recipients, with half of each cohort receiving immune checkpoint blockade (ICB; 5 mg/kg anti-PD-1) or PBS on days 7 and 14. Mice were monitored for tumor growth for 40 days. **C** The percent and number (inset) of mice with NC-derived and WD-derived sarcoma cell lines that developed a tumor of at least 500 mm³ is shown with

exact *P* values indicated for the bracketed groups calculated by two-sided Chi-squared analysis. **D** NC-derived and WD-derived sarcoma cell lines were injected subcutaneously into the flank of 10 secondary recipients, with half of each cohort depleted of T cells via injection with anti-Thy1.2 on days −1, 1, and 7. Tumor volume was tracked for 20 days and representative images of subcutaneous tumors are shown (all groups *n* = 10). *P* values were calculated by two-sided Mann–Whitney *U* test of tumor volumes at the day of euthanasia. **E** Survival of mice from *B* after anti-PD-1 treatment is displayed in a Kaplan–Meier graph with exact *P* value calculated via log-rank. **A** Created using BioRender.com.

To assess the durability of immune dysfunction in obesity, we investigated if weight loss through dietary changes could restore CD8+ TIL effector activity. When obese mice on WD were switched to a healthier low-fat diet, weight loss was accompanied by other improvements to general health including restored liver function and lowering of serum cholesterol. These previously obese animals also showed restored tumor immunity against B16 melanoma and improved responses to ICB immunotherapy when compared to obese mice maintained on WD. These results demonstrate that immune dysfunction associated with obesity is not necessarily permanent, and that diet-induced weight loss with subsequent improvements in metabolic health are sufficient to rescue T cell responses in the tumor microenvironment. Importantly, this scenario is distinct from cancer-

associated cachexia in patients, where dramatic loss of body mass coincides with systemic metabolic imbalance and lower clinical responses to immunotherapy[31]. Instead, our data provide evidence that dietary intervention and weight loss by individuals with obesity may be an avenue to improve antitumor immunity and treatment outcomes.

Advances in the development of obesity-fighting medications like semaglutide offer new hope for weight loss and have delivered promising results in clinical trials[23,24]. Therefore, we sought to determine if immune dysfunction in obese mice could be restored when weight loss was achieved through treatment with semaglutide, without changes in diet. Semaglutide was effective at inducing weight loss in obese mice that remained on WD, and treatment corresponded with a

normalization of fat and lean body mass composition that was equivalent to lean control mice. However, unlike diet-induced weight loss, semaglutide did not significantly improve liver health or other metabolic comorbidities, and these mice still experienced CD8[+] TIL dysfunction and poor responses to immunotherapy, similar to obese mice not treated with semaglutide. These in vivo experiments demonstrate that different weight loss strategies can appear equally successful when only body mass is taken into account yet have divergent impacts on the immune response to cancer.

The epidemiology linking obesity and cancer risk are clear, but whether compromised T cell immune responses contribute to this risk has not been established. In the B16 melanoma model, lean and obese mice displayed clear differences in immunoediting, resulting in an altered tumor phenotype. We predicted that such disruption of the immune surveillance processes would result in higher rates of tumor outgrowth in obese mice, but this could not be tested in a transplantable tumor model, thus necessitating the use of a chemical carcinogen-induced cancer model. Upon challenge with the MCA carcinogen, more than 84% of obese mice developed sarcoma tumors within 120 days, compared to only 53% of lean mice. This disparity was absent in lean and obese Rag2[−/−] mice that lack an adaptive immune system, suggesting that the protection afforded to lean mice was dependent on a robust antitumor immune response that was dampened in obese mice. These results provide experimental evidence linking obesity-related immune dysfunction with greater cancer risk. The fact that tumors developed at equivalent rates in lean and obese Rag2[−/−] mice, and similarly to obese wildtype mice, also indicates that impaired immune surveillance plays an oversized role in determining cancer risk compared to other obesity-associated comorbidities.

In cancer patients with obesity, the combination of impaired immune surveillance and less efficient immunoediting could impact therapeutic outcomes in unexpected ways. For example, the obesity paradox is based on the prediction that cancer patients with obesity would respond poorly to immunotherapy. Instead, survival outcomes of patients stratified by BMI are often similar and sometimes even better than non-obese patients when treated with ICB immunotherapy[15–18]. The mechanisms underlying this counterintuitive phenomenon have not been fully explained, but recent clinical findings indicate that differences in tumor mutational burden (TMB) play a role[43]. Seemingly conflicting data from another study suggested that human melanoma TMB is similar between lean and obese patients[44]. However, that analysis combined over-weight and obese cohorts (BMI > 25), hindering the ability to assess how TMB is influenced specifically in patients with obesity. Our data strongly support the notion that tumors undergo altered editing in obese hosts due to functional defects in CD8[+] TIL, and the resulting loss of vigorous immune surveillance places obese mice at greater risk of carcinogen-induced sarcomas. Extrapolation of these findings predicts that tumors developing in some individuals with obesity may be less edited and more immunogenic as a result, thus making them better targets once T cells are reinvigorated during immunotherapy. This was not recapitulated in the transferred B16 tumor model, where obese mice with melanoma responded poorly to ICB (Figs. 2 and 3). However, B16 tumors are derived from immune competent mice and have already been edited, which is a key distinction from the MCA sarcomas that developed under less immune pressure in the obese environment. Indeed, sarcoma tumor lines derived from obese mice were more often rejected or grew slower compared to sarcoma lines derived from lean mice following transfer into secondary healthy recipients. Sarcoma lines derived from lean and obese mice also differed in their sensitivity to ICB immunotherapy, as those from obese hosts were rejected in 75% of recipients treated with anti-PD-1, whereas tumors from lean mice were rejected in only 25% of recipients. We interpret these data as evidence that tumors progressing in the obese immune landscape endure less selective pressure, resulting in more immunogenic tumors that are effectively opposed by enhanced antitumor immune responses such as those generated during immunotherapy. This paradigm could explain the obesity paradox observed in some human cancer patients.

In summary, we used a mouse model of diet-induced obesity to investigate the utility of weight loss for restoring antitumor immunity. Our results demonstrate that immune dysfunction is regulated independently of body mass and adiposity and could not be restored by weight loss alone when induced by semaglutide treatment, but required more substantive improvements in diet and metabolic health. Moreover, immune dysfunction compromised immune surveillance and placed obese animals at greater risk of developing cancer, as T cells failed to efficiently clear potentially malignant cells prior to tumor formation. As a result, impaired tumor immunity also led to less efficient tumor editing, causing outgrown tumors from obese hosts to be more immunogenic and more responsive to immune checkpoint blockade therapy.

## Methods

### Mice

All mice were housed under pathogen-free conditions in the Saint Louis University School of Medicine Department of Comparative Medicine and used in accordance with animal use protocols approved by the Institutional Animal Care and Use Committee. C57BL/6 (Strain No. 000664), OT-II (Strain No. 004194) and Rag2[−/−] (Strain No. 033526) mice were purchased from the Jackson Laboratory. Mice were housed under a 12-h dark/light cycle, and housing was maintained at an ambient temperature of 72° Fahrenheit. Mice were age-matched and sex-matched and between 2 and 10 months of age when used for experiments. Mice were randomly assigned to either a normal chow (NC) diet with 21% kcal from fat and 23% kcal from protein (Lab Diet; cat. #0047039) or western diet (WD) chow containing 40% kcal from fat and 20% kcal from protein plus added sucrose (Research Diets; cat. #D19021301), and NC and WD mice were maintained in different cages in the same animal facility room until experimentation. Atorvastatin was provided in a custom WD chow developed by Research Diets containing 0.05% (500 mg/kg) atorvastatin sourced from Millipore Sigma (cat. #1044516). Mice were placed on NC or WD upon weaning at 4 weeks of age and maintained on the assigned diet until experimentation. Semaglutide-treated mice were maintained on WD for 12 weeks and then began biweekly treatment with intraperitoneal injections of 0.1 mg/kg semaglutide for 6 weeks. Percentages of lean and fat body mass were determined by nuclear magnetic resonance using a Bruker mini spec LF50. The mini spec acquired and analyzed time-domain nuclear magnetic resonance and provided body composition results for body mass of lean, fat, and fluid in each individual mouse. Percent fat mass was calculated by dividing the fat mass from total body mass and was reported as a percentage.

### Tumor immunotherapy

The B16-F0 (B16 hereafter) and YUMM melanoma tumor cell lines were purchased from the American Tissue Culture Collection (ATCC) (cat. #CRL-6322, cat. #CRL-3362). The B16 cell line was authenticated via short tandem repeat (STR) profiling and confirmed negative for *Mycoplasma* on February 2, 2020 by LabCorp Genetica Cell Line Testing (Burlington NC, USA). Cell lines other than the B16 cell line were not confirmed mycoplasma free. Cell lines were cultured in Dulbecco's Modified Eagle medium (DMEM; Thermo Fisher cat. #11995-065) with 10% fetal bovine serum (Corning; cat. #35-011-CV) and 1% penicillin-streptomycin (Sigma-Aldrich cat. #P0781). Adherent cells were removed with 0.25% trypsin, and cells were split 3–5 times for each experiment. For B16 tumor studies, $1 \times 10^6$ cells were injected subcutaneously into the flank of C57BL/6 mice. Tumors were established for 5-8 days prior to treatment with anti-PD-1 (clone RMP1-14; cat. #BE0146) and anti-CTLA-4 (clone 9D9; cat. #BE0164) obtained from Bio X Cell and administered intraperitoneally at 5 mg/kg on days

6 and 10 after tumor cell injection. For in vivo IFNγ neutralization, mice received intraperitoneal injection of 200 µg of anti-IFNγ (Bio X Cell; clone XMG1.2; cat. #BE0055) on days -1, 2, 5, 8, 11, and 14 relative to establishment of B16 tumors on day 0. Tumors were measured using digital calipers and tumor volume was determined using the equation ($L \times W^2$). For TIL analysis, mice were euthanized using carbon dioxide according to AVMA guidelines, and tumors were excised, mechanically disrupted with a sterile 3-mL syringe plunger and filtered through a 40-µm cell strainer. Isolation steps were performed in cold PBS.

## Flow cytometry

Fluorochrome-conjugated antibodies were purchased from Biolegend (anti-CD45, cat. #157612, clone QA17A26, host mouse, reactivity mouse, verified for flow cytometry, used for flow cytometry, dilution 1:200; anti-CD8a, cat. #100714, clone 53-6.7, host rat, reactivity mouse, verified for flow cytometry, used for flow cytometry, dilution 1:200; anti-CD4, cat. #100546, clone RM4-5, host rat, reactivity mouse, verified for flow cytometry, used for flow cytometry, dilution 1:200; anti-H-2Kb/H-2Db, cat. #114606, clone 28-8-6, host mouse, reactivity mouse, verified for flow cytometry, used for flow cytometry, dilution 1:200; anti-PD-1, cat. #135231, clone 29F.1A12, host rat, reactivity mouse, verified for flow cytometry, used for flow cytometry, dilution 1:200; anti-TIM3, cat. #119721, clone RMT3-23, host rat, reactivity mouse, verified for flow cytometry, used for flow cytometry, dilution 1:200; anti-LAG-3, cat. #125208, clone C9B7W, host rat, reactivity mouse, verified for flow cytometry, used for flow cytometry, dilution 1:200; anti-PECAM-1, cat. #102507, clone MEC13.3, host rat, reactivity mouse, verified for flow cytometry, used for flow cytometry, dilution 1:200; anti-ICAM-1, cat. #116121, clone YN1.7.4, host rat, reactivity mouse, verified for flow cytometry, used for flow cytometry, dilution 1:200; Granzyme B, cat. #515403, clone GB11, host mouse, reactivity human/mouse, verified for intracellular flow cytometry, used for intracellular flow cytometry, dilution 1:100; Perforin, cat. #154304, clone S16009A, host rat, reactivity mouse, verified for intracellular flow cytometry, used for intracellular flow cytometry, dilution 1:100; anti-IFNγ, cat. #505808, clone XMG1.2, host rat, reactivity mouse, verified for intracellular flow cytometry, used for intracellular flow cytometry, dilution 1:100), Thermo Fisher Scientific (anti-NK1.1, cat. #61-5941-82, clone PK136, host mouse, reactivity mouse, verified for flow cytometry, used for flow cytometry, dilution 1:200), BD Biosciences (anti-MHC-II I-a[b], cat. #562928, clone AF6-120.1, host mouse, reactivity mouse, verified for flow cytometry, used for flow cytometry, dilution 1:200), Invitrogen (anti-IFNγR, cat. #12-1191-82, clone 2E2, host Armenian hamster, reactivity mouse, verified for flow cytometry, used for flow cytometry, dilution 1:200), and Abcam (Gp100, cat. #246730, clone EP4863(2), host rabbit, reactivity mouse, verified for intracellular flow cytometry, used for intracellular flow cytometry, dilution 1:100). All flow cytometry verified antibodies were used at a 1:200 dilution and all intracellular flow cytometry verified antibodies were used at a 1:100 dilution. Tyrosinase-related peptide-2 (TRP-2) tetramer was obtained from the National Institutes of Health Tetramer Core Facility (H-2K$^b$ amino acid sequence SVYDFFVWL in color BV-421). TRP-2 tetramer was used at a 1:200 dilution. Aqua florescent reactive dye (live/dead) was purchased from Invitrogen (cat. #L34966A) and used at a 1:1000 dilution, and Fc block was purchased from Biolegend (cat. #101320) and used at a 1:200 dilution. To induce MHC-I/II expression on melanoma tumor cells, the B16 line was incubated with 0, 1, or 10 U/mL of recombinant mouse IFNγ (Roche Diagnostics; cat. #11276905001) for 24 h. Intracellular staining of cytoplasmic- and nuclear-associated proteins was performed using the eBioscience cellular permeabilization kit (cat. #00-5523-00) per the manufacturer's instructions. Briefly, cells were processed and stained ex vivo with live/dead and cell surface markers described above. Cells were then fixed, permeabilized, and stained with antibodies specific for the intracellular proteins Granzyme B (Biolegend; cat. #515403, clone GB11, host mouse, reactivity human/

mouse, verified for intracellular flow cytometry, used for intracellular flow cytometry, dilution 1:100), Perforin (Biolegend; cat. #154304, clone S16009A, host rat, reactivity mouse, verified for intracellular flow cytometry, used for intracellular flow cytometry, dilution 1:100), or Gp100 (Abcam; cat. #246730, clone EP4863(2), host rabbit, reactivity mouse, verified for intracellular flow cytometry, used for intracellular flow cytometry, dilution 1:100). Intracellular cytokine staining was performed using the Cytofix/Cytoperm plus kit (BD Biosciences) per the manufacturer's instructions. Briefly, cells were incubated ex vivo with 10 ng/mL PMA (Sigma-Aldrich; cat. #P8139) and 0.3 µg Ionomycin (Sigma-Aldrich; cat. #I0634) for 4 h in the presence of GolgiPlug (BD Biosciences; 51-2301KZ). Cells were first stained with live-dead and cell surface markers as described above, then fixed, permeabilized, and stained with anti-IFNγ (Biolegend; cat. #505808, clone XMG1.2, host rat, reactivity mouse, verified for intracellular flow cytometry, used for intracellular flow cytometry, dilution 1:100). Flow-cytometric analysis was performed on LSRFortessa Cell Analyzer (BD Biosciences) in the Saint Louis University Flow Cytometry Core Facility, and data analyzed using FlowJo v.10 software (Tree Star Inc.).

## Carcinogen-induced tumors

Male C57BL/6 and Rag2$^{-/-}$ mice were maintained on NC and WD for 12 weeks prior to a single subcutaneous injection of 50 µg of 3′-methylcholanthrene (MCA) (Millipore Sigma; cat. #200-276-4) dissolved in corn oil. Tumor volumes were monitored for 120 days. Primary endpoint of this study was tumor incidence, defined as time at which tumors reached 8 mm in diameter. For sarcoma tumor transfer studies, cell lines originally derived from MCA-treated mice were reimplanted into 8-week-old C57BL/6 mice and tumor volumes were monitored for 40 days. In vivo T cell depletion was achieved by IP injection of 200 µg of anti-Thy1.2 (30H12) from Bio X Cell (cat. #BE0164) on days -1, 1, and 7 relative to tumor cell injections.

## Serum chemistry analysis

Mouse blood samples were collected via cheek bleed and serum was isolated via centrifugation. Serum was analyzed by Midwest Vet Labs using a Beckman Coulter AU480 system (Roche) to quantify total cholesterol and the liver enzymes alanine transaminase (ALT) and aspartate aminotransferase (AST) using reagents from Sekisui Diagnostics according to manufacturer's protocols. Glucose and insulin levels were measured in the serum of mice fasted for 3 h using a Contour Next blood glucose monitoring system and an Ultra Sensitive Mouse Insulin ELISA Kit (Crystal Chem, cat. #90080).

## Histopathologic assessment

Mouse organs were harvested and embedded in paraffin. Tissues were sectioned, slides were created in the Saint Louis University Microscopy Core Facility, and the slides were stained with hematoxylin and eosin.

## Single-cell RNA sequencing

CD8$^+$ TIL were enriched from tumors using CD8$^+$ MicroBeads (Miltenyi Biotec; cat. #130-116-478) and separated on LS columns (Miltenyi Biotec; cat. #130-042-401) using a QuadroMACS Cell Separator (Miltenyi Biotec, cat. #130-090-976). Enriched TIL were then sorted to at least 99% purity based on CD8 staining using a BD FACSAria III (BD Biosciences), and single-cell suspensions were loaded on a Chromium Single-Cell Controller instrument (10X Genomics) to generate single-cell gel beads in emulsion (GEMs). Single-cell RNA libraries were prepared using the Chromium Single-Cell 3′ Library and Gel Bead Kit PN-1000268. GEM-RT was performed in a T100 96-Well Thermal cycler (Applied Biosystems, 4375786) according to 10X protocol. The GEMs were then broken, and single-stranded cDNA was cleaned up with DynaBeads MyOne Silane Beads (P/N 2000048). Barcoded, full-length cDNA was amplified using the T100 96-Well Thermal Cycler according to 10X protocol and stored at −20 °C. Amplified cDNA product was

cleaned up with the SPRIselect Reagent Kit (0.6 X SPRI; Beckman Coulter; P/N B23318). 3′ gene-expression libraries were constructed using the reagents from the Chromium Single-Cell 3′ Library Construction Kit (P/N 1000190). Quality control and Illumina sequencing of the libraries was performed on the NovaSeq6000 platform at the Washington University in St. Louis Genome Technology Access Center (GTAC). Raw data were processed through the CellRanger 6.0.2 pipeline (10x Genomics), and clustering and differential expression analysis was conducted with the open-source R v2023.09.1 + 494 software package Seurat v4.3.0. Cells were filtered to retain only those with less than 7.5% mitochondrial RNA, and unique molecular identifiers (UMIs) > 500 and <50,000. After sorting, processing, and QC, 5,320 total CD8$^+$ T cells were analyzed (2697 from NC, and 2623 from WD). Visualization of UMAP and heatmaps were generated using ggplot2 v3.4.3. Unbiased Hallmark pathway analysis was conducted using the msigdbr v7.5.1 and SCPA v1.5.4 packages[45]. The single-cell RNA sequencing data generated in this study have been deposited in the Gene Expression Omnibus database under accession code GSE245657.

## Statistics and reproducibility

No statistical method was used to predetermine sample size. Mice were randomly assigned to either control or experimental groups when received from Jackson Labs. No data were excluded from the analysis and source data for all figures have been made available with the paper. The investigators were not blinded to allocation during the experiments and outcome assessment. Statistical analysis to compare treatment groups was performed using nonparametric, two-sided Mann–Whitney U tests (Prism 9.5.1, GraphPad Software). Linear regression analysis was performed via calculation of the Pearson correlation coefficient ($r$) and corresponding $P$ value using Prism 9.5.1. Tumor incidence and survival curve statistics were estimated using Kaplan-Meier methodology and compared via log-rank test using Prism 9.5.1 and post-hoc analysis was performed. Categorical tumor incidence was compared using two-sided, Chi-squared tests between mouse cohorts using Prism 9.5.1. Box-and-whisker plots: The box indicates the 25$^{th}$ and 75$^{th}$ percentile, the line indicates the data median, and the whiskers indicate the minimum and maximum of all individual values. Standard error of the mean is represented for all plots showing individual values. Exact $P$ values are indicated for all data.

## Reporting summary

Further information on research design is available in the Nature Portfolio Reporting Summary linked to this article.

## Data availability

The single-cell RNA sequencing data generated in this study have been deposited in the Gene Expression Omnibus database under accession code GSE245657. Source data for all figures have been provided with the paper. Source data are provided with this paper.

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

## Acknowledgements

This study was supported by grants from the National Institutes of Health National Cancer Institute (RO1 CA238705 and P30 CA091842) to R.M.T. The authors thank the Saint Louis University Flow Cytometry Core, Genomics Core, and Department of Comparative Medicine for their technical assistance.

## Author contributions

A.P. Conceptualization, Methodology, Formal Analysis, Validation, Investigation, Writing – Original Draft, Writing – Review and Editing, Visualization; E.E. Software, Validation, Investigation, Data Curation, Writing – Review and Editing; C.G. Investigation, Writing – Review and Editing, N.K. Methodology, Software, Investigation, Writing – Review and Editing; L.M.K. Conceptualization, Methodology, Writing – Review and Editing; S.G.H. Software, Investigation, Writing – Review and Editing; K.D.P Investigation, Validation, Formal Analysis; K.S.M. Resources, Supervision, Project Administration; R.J.D. Resources, Supervision, Writing – Review and Editing; S.T.F. Methodology, Supervision, Writing – Review and Editing; E.A. Conceptualization, Methodology, Resources, Writing – Review and Editing; R.M.T Conceptualization, Methodology, Validation, Formal Analysis, Investigation, Resources, Data Curation, Writing – Original Draft, Writing – Review and Editing, Visualization, Supervision, Project Administration, Funding Acquisition

## Competing interests

The authors declare no competing interests.
