## [Peer Review File · Nature Communications]

Obesity-related T cell dysfunction impairs immunosurveillance and increases cancer riskREVIEWER COMMENTS

Reviewer #1 (Remarks to the Author):

The manuscript by Piening et al. makes multiple important contributions to the field of obesity & immunotherapy research. For example, the comparisons of weight loss achieved via dietary restriction versus semaglutide to study effects on CD8+ T cell tumor immunity are timely, unique, and beautifully conducted. The Semaglutide experiments, in particular, are novel and yield important new information that will likely have application beyond anti-tumor immunity and cancer immunotherapy outcomes. Likewise, examining the effects of obesity-induced impairments in immunoediting and how that might influence tumor immunogenicity and response to immunotherapy is a thoughtful, nuanced approach to the question of how obesity impacts anti-tumor immunity and immunotherapy outcomes. The thoroughness and elegance of the models used in Figures 4D/E, Figure 5, and Figure 6 are outstanding. Throughout, the experimental methods used are well-controlled and carefully designed. This is one of the more insightful attempts to address the very complex issues surrounding obesity, weight loss, and tumor immunity. Thus, the significance of this manuscript is high.

However, there are a few loose ends that need to be tied in order for this manuscript to really shine.

Major and minor comments are below:

1. The authors conclude that weight loss is beneficial for anti-tumor immunity (Discussion). However, the authors fail to show what happens to the weights of their mice and their animals' lean mass, specifically, after tumor challenge in lean vs obese animals. Don't the obese mice lose weight? Please show these data if they are available. If weight loss occurs, wouldn't this naturally enhance immune function in these animals?
2. In a related point: The authors state "For some obese patients, weight loss through strict dietary changes may not be practical." (Discussion, Line 363) But weight loss in cancer patients is known to be detrimental and associated with poor outcomes. How do the authors reconcile their observations and conclusions with this well-known fact?
3. The authors conclude that "Extrapolation of these findings predicts that tumors developing in some obese people may be less edited and more immunogenic as a result, thus making them better targets for reinvigorated T cells elicited during immunotherapy." (Discussion) But their own experiments show that in obese mice with melanoma, ICB works less well. This would seem to be more relevant to the human condition versus the scenario in Figure 6 where tumors from obese mice were transplanted into lean mice then treated with ICB. How do the authors reconcile these differences?
4. Figure 4 is intriguing but not quite where it needs to be to convincingly demonstrate the authors' conclusions.
 - a. Gating on CD45- cells is insufficiently rigorous for the conclusions being made. What if there is an increase in the frequency of stromal cells in lean mice? Wouldn't this make the % of GP100+ cells lower?
 - b. Gp100/pmel 17 peptides antigens (different from the intracellular protein the authors are detecting by staining & flow) can be presented by MHC I and MHC II. See <https://doi.org/10.4049/jimmunol.181.11.7843> Thus, the authors should show ex vivo MHC I and MHC II staining on B16 tumor cells. Fibroblasts and endothelial cells could be excluded via a dump gate.
 - c. The vast majority of CD45- "tumor" cells in both lean and obese mice are Gp100-negative as per Fig. 4B, which suggests that immunoediting is actually occurring – even in obese mice – although the Gp100+ cells appear to express almost no MHC I. This leads back to the question of whether Gp100 peptide antigens are being expressed by MHC II. The authors should comment on

their interpretation of these results.

d. The authors appear to be suggesting with Figure 4E that in the absence of selective pressure in obese mice, melanoma cells do not express MHC I. So, is the lack of MHC I expression due to reduced production of IFN γ in the TME? If the authors block IFN γ in vivo in lean mice, will the phenotype of low tumor cell MHC I and high GP100 be reproduced? This is an important point of the manuscript that needs to be demonstrated more fully.

5. The authors can't conclude that "The inability to efficiently eliminate immunogenic Gp100+ melanoma cells suggests that compromised immunity in obese hosts impairs immune surveillance." (Lines 312-314) If the tumor cells never had MHC I, they couldn't be recognized by CD8+ T cells, and therefore would not be considered immunogenic. Please revise this statement.

Minor:

1. The staining protocol for GP100 should be clarified or referenced. Ex: was methanol used for fixation with detergent permeabilization or was the BD CytoFix/CytoPerm kit used for this as well as intracellular cytokines?

2. The authors need to make clear in lines 296-298 that the Gp100/ pmel being detected by antibody staining is an intracellular protein, which is distinct from Gp100-derived peptide antigens that can be presented by MHC I and II.

Reviewer #2 (Remarks to the Author):

Obesity is a risk factor for cancer. In this study, the authors aimed to address two important questions. 1) Can the impaired T cell response in obese mice be restored following weight loss through two different means (diet intervention vs. semaglutide treatment)? 2) Can the impaired immune response and immunoediting in obese mice make tumor grown in those mice more immunogenic? They designed a series of nice experiments to show that dietary restriction, but not semaglutide, could restore T cell function and improve response to immunotherapy. Furthermore, they found that impaired immunoediting in the obese environment enhanced tumor immunogenicity, making them highly sensitive to immunotherapy following transplantation into lean mice. Overall, this is a very nicely conducted research paper that bring important insights to the regulation of tumor immunity during obesity. However, there are a few issues that I hope the authors could address:

1. Fig 3. The mechanism by which dietary restriction but not semaglutide could restore T cell anti-tumor immunity is unclear. The authors mentioned that although both approaches lowered body weight and fat mass, only dietary restriction improved liver health and hyperlipidemia. Is the increased lipid content directly associated with impaired T cell response and tumor volume? They could perform some association studies to show correlations between lipid/cholesterol content and T cell response/tumor volume. They should also discuss other potential factors that may underline the differences between the two groups, such as microbiota differences, influence of Glp1 signaling on immune function...

2. Fig 4. Why do tumors grown in obese mice fail to upregulate MHC-I in response to IFN γ ? If it is not because of the levels of IFN γ R, could it be due to impaired IFN γ R signaling, such as impaired STAT1 phosphorylation? Wouldn't this make such tumors less immunogenic rather than more immunogenic? Maybe NK cells play a role here?

3. Fig 5 and 6. They should characterize the T cell response in the sarcoma model.

Reviewer #3 (Remarks to the Author):

Obesity and associated metabolic disorders have been extensively reported to increase risk for developing various cancers. As a follow-up of their previous work (Kuehm et al. *Cancer Immunol Res* 2021), Piening et al. explored further the role of obesity-induced T cell dysfunction as one of the possible underlying mechanisms in several pre-clinical mouse models of tumorigenesis. In the present manuscript, they confirmed that tumor-infiltrating CD8+ T cells (TIL) from high-fat diet (HFD)-fed obese mice present some characteristic of altered effector functions, a feature associated with impaired response to anti-PD-1/anti-CTLA-4 immunotherapy. Interestingly, they demonstrated that body weight loss in obese mice induced by shifting back to normal diet, but not by Semaglutide treatment, restores CD8+ TIL functions and improves tumor immunity. Finally, they showed that MCA-induced tumors from obese hosts are cleared more efficiently than those from lean hosts when implemented into lean mouse recipients, suggesting reduced T-cell mediated tumor immunoeediting in an obesogenic environment. Altogether, the manuscript is interesting and well-written, containing some novel findings supported by solid data obtained using relevant experimental approaches. Few points mentioned below would however require clarification.

Major comments

- The authors showed that MACS-sorted CD8+ TIL in B16 tumor from obese mice have reduced intracellular expression of Ifng/GzmB/Perforin, suggesting impaired effector function. Is there any possible approach, e.g. co-culture assay, that could be used to directly confirm a decrease in anti-tumor CD8+ T cell cytotoxicity ?
- Circulating levels of insulin, a pro-tumorigenic growth factor, are usually very significantly affected by HFD and in response to body weight loss. However, contrasting with what will be observed following diet-induced body weight loss, treatment with the GLP1 receptor agonist Semaglutide would presumably promote insulin secretion by beta cells (despite inducing brain-mediated inhibition of food intake), a feature that may worsen HFD-induced hyperinsulinemia and contribute to the observed differences in tumor immunogenicity in these 2 models. It would therefore be relevant to measure this serum parameter and at least discuss the putative contribution of hyperinsulinemia in T cell dysfunction (e.g. Fisher et al. *J Immunol* 2017; Tsai et al *Cell Metabolism* 2018). Similarly, it is surprising that, by contrast to cholesterol, serum glucose levels, a key nutrient for both tumor and T cells, were not determined. Taken into account the central role of T cell-intrinsic metabolism in shaping their effector functions, one may question whether any cluster-specific differences in transcriptional 'metabolic' signature can be found in tumor-sorted CD8+ TIL from lean and obese mice. As such, a non-biased gene ontology analysis of the scRNAseq data would be useful, at least as supplementary data.

Minor

- Please provide the composition/reference of normal chow diet
- Line 228: "increased serum liver enzymes", used as readout for HFD-induced liver damage
- Figure 3G: p-values not aligned
- Figure 4D-E: color coding for lean OT-II mice may be confusing, with the same one used in panel A-C for WD-fed obese mice.

Reviewer #1 (Remarks to the Author):

The manuscript by Piening et al. makes multiple important contributions to the field of obesity & immunotherapy research. For example, the comparisons of weight loss achieved via dietary restriction versus semaglutide to study effects on CD8+ T cell tumor immunity are timely, unique, and beautifully conducted. The Semaglutide experiments, in particular, are novel and yield important new information that will likely have application beyond anti-tumor immunity and cancer immunotherapy outcomes. Likewise, examining the effects of obesity-induced impairments in immunoediting and how that might influence tumor immunogenicity and response to immunotherapy is a thoughtful, nuanced approach to the question of how obesity impacts anti-tumor immunity and immunotherapy outcomes. The thoroughness and elegance of the models used in Figures 4D/E, Figure 5, and Figure 6 are outstanding. Throughout, the experimental methods used are well-controlled and carefully designed. This is one of the more insightful attempts to address the very complex issues surrounding obesity, weight loss, and tumor immunity. Thus, the significance of this manuscript is high.

However, there are a few loose ends that need to be tied in order for this manuscript to really shine.

Major and minor comments are below:

1. The authors conclude that weight loss is beneficial for anti-tumor immunity (Discussion). However, the authors fail to show what happens to the weights of their mice and their animals' lean mass, specifically, after tumor challenge in lean vs obese animals. Don't the obese mice lose weight? Please show these data if they are available. If weight loss occurs, wouldn't this naturally enhance immune function in these animals?

This is an important point to clarify and we have added new data and commentary to address this issue. Our conclusions have not changed, as our original data support that weight-loss alone is insufficient to improve tumor immunity, as demonstrated in mice treated with semaglutide (**Fig. 3**). However, improvements in overall health that coincided with diet-induced weight-loss were beneficial (**Fig. 2**). To specifically address the reviewer's question, we have provided new data documenting that body mass is maintained in both lean and obese mice over the course of a 15-day in vivo tumor challenge study (**Suppl. Fig. S4**).

2. In a related point: The authors state "For some obese patients, weight loss through strict dietary changes may not be practical." (Discussion, Line 363) But weight loss in cancer patients is known to be detrimental and associated with poor outcomes. How do the authors reconcile their observations and conclusions with this well-known fact?

We agree with the reviewer that cancer patients can lose body weight (cachexia) and skeletal muscle mass (sarcopenia), and these patients often experience worse immunotherapy outcomes (reviewed in PMID: 31389842). However, weight loss was not observed during the 15-day time course of our transplanted B16 tumor studies (**Suppl. Fig. S4**). Still, the clarity of language to this point was lacking in our original manuscript and we have taken this opportunity to provide additional commentary and relevant citations in order to differentiate between weight-loss in obese patients through healthier diet versus the detrimental effects of cancer-induced cachexia. This new data is described on line 259 in the Results section and line 408 on the Discussion in the revised manuscript.

3. The authors conclude that "Extrapolation of these findings predicts that tumors developing in some

obese people may be less edited and more immunogenic as a result, thus making them better targets for reinvigorated T cells elicited during immunotherapy.” (Discussion) But their own experiments show that in obese mice with melanoma, ICB works less well. This would seem to be more relevant to the human condition versus the scenario in Figure 6 where tumors from obese mice were transplanted into lean mice then treated with ICB. How do the authors reconcile these differences?

This is an important and complex issue that deserves more clarification in the manuscript. The statement above applies to tumors arising in the obese environment where immune surveillance is compromised during all stages of tumor development, but this was not made sufficiently clear in the original manuscript. The reviewer correctly points out that in the transplanted B16 melanoma model, ICB worked less well in obese mice (**Fig. 2-3**) but B16 tumors were originally derived from immune competent mice, and thus are expected to have already been edited and escaped immune-mediated killing. Thus, even when impaired immune responses in the obese mice are modestly boosted with ICB, these tumors are still not well controlled. This scenario is distinct from de novo tumors that arise over several months in obese mice challenged with MCA, which are continually shaped by the immune system. Our strategy to transfer MCA-induced tumors into secondary recipients (**Fig. 6**) enabled us to directly compare the immunogenicity of sarcomas derived from lean versus obese mice in a controlled in vivo setting. The results of these studies were unambiguous, as MCA-tumors from obese mice were consistently better controlled and were rejected more frequently than tumors derived from lean mice. This was observed in untreated secondary recipients and enhanced in those receiving ICB immunotherapy. We interpret this as evidence that tumors arising under less immune pressure in obese hosts are more immunogenic, leading us to the prediction stated above, which is admittedly provocative but still supported by these preclinical studies. We have revised the manuscript to provide new data on tumor immunogenicity (**Suppl. Fig. S10**) and to provide more clarity on this issue in the Discussion (line 451).

4. Figure 4 is intriguing but not quite where it needs to be to convincingly demonstrate the authors' conclusions.

a. Gating on CD45- cells is insufficiently rigorous for the conclusions being made. What if there is an increase in the frequency of stromal cells in lean mice? Wouldn't this make the % of Gp100+ cells lower?

We agree with the reviewer that results could be misleading if lean and obese mice have differences in stromal cells within tumors, leading to inaccurate calculations of the frequency of Gp100+ cells when gating only on CD45-negative cells. As suggested by the reviewer, we have improved our gating strategy to minimize analysis of stromal cells using antibodies specific for PECAM-1 and ICAM-1 in a dump gate, and have provided a new figure showing this strategy (**Suppl. Fig. S6**). Experimental results using this more rigorous approach did not change the overall conclusion of the study, as the frequency of Gp100+ cells within melanoma tumors was still significantly higher in obese compared to lean mice (**Fig. 4B**).

b. Gp100/pmel 17 peptides antigens (different from the intracellular protein the authors are detecting by staining & flow) can be presented by MHC I and MHC II.

See <https://doi.org/10.4049/jimmunol.181.11.7843> Thus, the authors should show ex vivo MHC I and MHC II staining on B16 tumor cells. Fibroblasts and endothelial cells could be excluded via a dump gate.

c. The vast majority of CD45- “tumor” cells in both lean and obese mice are Gp100-negative as per Fig. 4B, which suggests that immunoediting is actually occurring – even in obese mice – although the Gp100+ cells appear to express almost no MHC I. This leads back to the question of whether Gp100

peptide antigens are being expressed by MHC II. The authors should comment on their interpretation of these results.

The reviewer correctly points out that the staining performed in our analysis identifies intracellular Gp100 protein, which is distinct from peptide antigens presented via either MHC-I or MHC-II. We have provided additional clarity on this point in the revised Results section (line 306). As suggested by the reviewer, we have included new data showing ex vivo MHC-I and MHC-II expression on Gp100+ melanoma tumor cells (**Fig. 4**), employing a more rigorous gating strategy that excludes assessment of fibroblast and endothelial cell populations (**Suppl. Fig. S6**). Results from this new analysis improved the quality of the data, and confirmed a decrease in MHC-I expression on Gp100+ tumor cells from obese mice, as well as a modest reduction of MHC-II expression.

d. The authors appear to be suggesting with Figure 4E that in the absence of selective pressure in obese mice, melanoma cells do not express MHC I. So, is the lack of MHC I expression due to reduced production of IFN γ in the TME? If the authors block IFN γ in vivo in lean mice, will the phenotype of low tumor cell MHC I and high GP100 be reproduced? This is an important point of the manuscript that needs to be demonstrated more fully.

Indeed, the interconnected role between IFN γ and tumoral MHC-I expression is an important point of the manuscript. As suggested by the reviewer, we have now performed new in vivo IFN γ neutralization experiments (**Suppl. Fig. S7**), demonstrating that blocking IFN γ reduces MHC-I and MHC-II expression on tumors. However, there was some disconnect between expression of MHC and Gp100, as the frequency of Gp100+ cells was unaffected by IFN γ neutralization, which could indicate unanticipated effects of reducing IFN γ on the broader anti-tumor response, or a limit to what can be achieved by this approach. Perspective on these data and how they fit conceptually into our study is provided in the revised Results section (line 324).

5. The authors can't conclude that "The inability to efficiently eliminate immunogenic Gp100+ melanoma cells suggests that compromised immunity in obese hosts impairs immune surveillance." (Lines 312-314) If the tumor cells never had MHC I, they couldn't be recognized by CD8+ T cells, and therefore would not be considered immunogenic. Please revise this statement.

We agree with the reviewer, and have now edited the sentence to say, "The inability to efficiently eliminate ~~immunogenic~~ Gp100+ melanoma cells suggests that compromised immunity in obese hosts ~~impairs~~ alters immune surveillance," in the revised manuscript (line 343).

Minor:

1. The staining protocol for GP100 should be clarified or referenced. Ex: was methanol used for fixation with detergent permeabilization or was the BD CytoFix/CytoPerm kit used for this as well as intracellular cytokines?

We have updated the Methods section to include a more complete description of the reagents and protocols used fix and permeabilize cells for detection of intracellular GP100 protein and cytokines (line 139).

2. The authors need to make clear in lines 296-298 that the Gp100/ p μ mel being detected by antibody

staining is an intracellular protein, which is distinct from Gp100-derived peptide antigens that can be presented by MHC I and II.

We agree this is an important distinction, and have edited the Results section to more clearly describe the Gp100 being detected (line 306).

Reviewer #2 (Remarks to the Author):

Obesity is a risk factor for cancer. In this study, the authors aimed to address two important questions. 1) Can the impaired T cell response in obese mice be restored following weight loss through two different means (diet intervention vs. semaglutide treatment)? 2) Can the impaired immune response and immunoediting in obese mice make tumor grown in those mice more immunogenic? They designed a series of nice experiments to show that dietary restriction, but not semaglutide, could restore T cell function and improve response to immunotherapy. Furthermore, they found that impaired immunoediting in the obese environment enhanced tumor immunogenicity, making them highly sensitive to immunotherapy following transplantation into lean mice. Overall, this is a very nicely conducted research paper that bring important insights to the regulation of tumor immunity during obesity. However, there are a few issues that I hope the authors could address:

1. Fig 3. The mechanism by which dietary restriction but not semaglutide could restore T cell anti-tumor immunity is unclear. The authors mentioned that although both approaches lowered body weight and fat mass, only dietary restriction improved liver health and hyperlipidemia. Is the increased lipid content directly associated with impaired T cell response and tumor volume? They could perform some association studies to show correlations between lipid/cholesterol content and T cell response/tumor volume. They should also discuss other potential factors that may underline the differences between the two groups, such as microbiota differences, influence of Glp1 signaling on immune function...

We share the reviewer's interest in identifying the obesity-related comorbidities that directly influence T cell function. Based on our previous studies (Khojandi et al 2021), we also suspected that hyperlipidemia may play a role here. As suggested by the reviewer, we have added new association data (**Suppl. Fig. S5**), which unexpectedly showed no correlation between serum cholesterol and CD8+ TIL function or tumor volume in obese mice. Despite this negative result, we pursued this further in a series of new in vivo experiments where obese mice on WD were treated with atorvastatin to reduce cholesterol. Although serum cholesterol levels were successfully lowered, this did not significantly improve CD8+ TIL function (**Suppl. Fig. S5**), suggesting that hyperlipidemia does not explain the immune defects associated with obesity in these mice. This new data is described in the revised Results section (line 284). We previously reported difference in the microbiomes of lean versus obese mice, but fecal transfer studies suggested this did not contribute to immune dysfunction (Kuehm et al, 2021). The gene ontology analysis we have now conducted (**Suppl. Fig. S2**) has provided some new perspective into the metabolic pathways that distinguish TIL from lean and obese mice. In particular, CD8+ TIL from obese mice displayed a transcriptional signature consistent with oxidative phosphorylation, suggesting that they failed to switch to aerobic glycolysis. This switch is thought to be required for gain of effector function, thus these results provide new insight into a possible mechanism of obesity-related immune dysfunction, as discussed in the revised manuscript (line 235). Although we have not identified the obesity-associated factors that dictate this altered metabolism, this gap in knowledge does not detract from the overall impact

of our study that demonstrates for the first time that impaired immune surveillance underlies the increased risk of cancer associated with obesity.

2. Fig 4. Why do tumors grown in obese mice fail to upregulate MHC-I in response to IFN γ ? If it is not because of the levels of IFN γ R, could it be due to impaired IFN γ R signaling, such as impaired STAT1 phosphorylation? Wouldn't this make such tumors less immunogenic rather than more immunogenic? Maybe NK cells play a role here?

This is an important point, and similar to issues raised by other reviewers. As a clarification to the reviewer's comment, our data suggest that tumors in obese mice fail to upregulate MHC-I because of low IFN γ production from dysfunctional CD8+ TIL. This was observed at both the gene expression and protein level (**Figs. 1 and 2**). Thus, these tumors are not exposed to IFN γ . To address this experimentally, we have now performed new in vivo IFN γ neutralization studies, demonstrating that blocking IFN γ in lean mice results in lower MHC-I and MHC-II expression on tumors, and this data is now provided in the revised manuscript (**Suppl. Fig. S7**). However, as the reviewer astutely points out, lower MHC-I expression by tumors would reduce their immunogenicity and this raises a "chicken and egg" scenario. In other words, does low IFN γ expression by dysfunctional T cells fails to induce MHC-I on tumors, or does low MHC-I expression on tumor cells reduce their immunogenicity such that T cells do not get activated? Our data support the first scenario, where low IFN γ expression by dysfunctional T cells in obese mice results in the failure to induce MHC expression on tumor cells. This is based on several observations indicating that CD8+ TIL in obese mice are indeed responding to antigen stimulation. First, tumors from lean and obese mice have similar frequencies of infiltrating T cells, and this is true of total CD8+ TIL and those specific for the TRP-2 tumor-associated antigen (**Suppl. Fig. S3**). Second, these TIL populations expanded to similar levels after ICB treatment (**Suppl. Fig. S3**). Additionally, other indicators of activation that are not directly related to cytolytic function, such as *Mki67* (Ki67), *Tbx21* (Tbet), and *Pdcd1* (PD-1) were all equivalently expressed in CD8+ TIL from both lean and obese mice (**Fig. 1**). We interpret these results to suggest that a lack of antigen detection within B16 tumors is not a likely explanation for impaired effector function by CD8+ TIL in the obese mice. We have now provided more commentary on this complex issue in the revised Results section (line 334). As for the possible role of NK cells, we observe equivalent frequencies and function in tumors from both lean and obese mice, and have now provide this data in a new figure (**Suppl. Fig. S8**) in the revised manuscript. Although not exhaustive, these data do not suggest any disparities in NK cell function between lean and obese mice.

3. Fig 5 and 6. They should characterize the T cell response in the sarcoma model.

We agree with this suggestion and have now performed new in vivo experiments to characterize the T cell responses to sarcoma tumors derived from lean versus obese mice following transplantation into secondary recipients. Results showed clear increases in CD8+ TIL responses to tumors from obese compared to lean mice based on expression of Ki67, Tbet, GzmB and Perforin, and these data are now provided in a new figure in the revised manuscript (**Suppl. Fig. S10**). This outcome strengthens our original conclusion that tumors derived from obese mice are more immunogenic than those from lean mice.

Reviewer #3 (Remarks to the Author):

Obesity and associated metabolic disorders have been extensively reported to increase risk for developing various cancers. As a follow-up of their previous work (Kuehm et al. Cancer Immunol Res

2021), Piening et al. explored further the role of obesity-induced T cell dysfunction as one of the possible underlying mechanisms in several pre-clinical mouse models of tumorigenesis. In the present manuscript, they confirmed that tumor-infiltrating CD8+ T cells (TIL) from high-fat diet (HFD)-fed obese mice present some characteristic of altered effector functions, a feature associated with impaired response to anti-PD-1/anti-CTLA-4 immunotherapy. Interestingly, they demonstrated that body weight loss in obese mice induced by shifting back to normal diet, but not by Semaglutide treatment, restores CD8+ TIL functions and improves tumor immunity. Finally, they showed that MCA-induced tumors from obese hosts are cleared more efficiently than those from lean hosts when implemented into lean mouse recipients, suggesting reduced T-cell mediated tumor immunoediting in an obesogenic environment. Altogether, the manuscript is interesting and well-written, containing some novel findings supported by solid data obtained using relevant experimental approaches. Few points mentioned below would however require clarification.

Major comments

- The authors showed that MACS-sorted CD8+ TIL in B16 tumor from obese mice have reduced intracellular expression of Ifng/GzmB/Perforin, suggesting impaired effector function. Is there any possible approach, e.g. co-culture assay, that could be used to directly confirm a decrease in anti-tumor CD8+ T cell cytotoxicity?

We wholeheartedly agree that an *ex vivo* co-culture assay would be ideal for testing cytotoxicity. However, after much consideration, we have determined that the logistical barriers of such an approach unfortunately make it practically impossible here. First, experience from our scRNAseq studies (**Fig. 1**) informed us that to obtain a purified population of just 100,000 CD8+ TIL, tumors from approximately 10 mice per experimental group would need to be processed, then enriched for T cells by MACS-sorting, and then undergo several hours of cell sorting. Although this is technically achievable, it is not clear whether this number of heterogeneous TIL would even be sufficient to setup a well-controlled assay with necessary replicates. Even with 100,000 polyclonal endogenous CD8+ TIL, it is expected that many (perhaps a majority) will be bystander T cells unable to target tumor-antigens. There are also questions about whether T cells that undergo such extensive *ex vivo* processing will retain sufficient function to be accurately detected, especially for heterogeneous TIL in such low numbers. In contrast, we have clear and consistent data demonstrating that CD8+ TIL from obese mice have significantly lower expression of GzmB, Perforin and IFNg at both the gene (**Fig. 1**) and protein (**Fig. 2-3**) levels. Admittedly these are only surrogate markers for cytolytic function, but the establish role of these molecules for CD8-mediated killing has made their detection the accepted standard for experimentally measuring effector function, particularly when all three are assessed at once. Thus, even if completely successful, the co-culture studies discussed above would provide only confirmation of what is already understood. Given the considerable costs associated with these co-culture assays (e.g. time, funding, and mice), which would require several experiments dedicated just to assay development followed by multiple replicate experiments, their potential for impact is expected to be relatively modest.

- Circulating levels of insulin, a pro-tumorigenic growth factor, are usually very significantly affected by HFD and in response to body weight loss. However, contrasting with what will be observed following diet-induced body weight loss, treatment with the GLP1 receptor agonist Semaglutide would presumably promote insulin secretion by beta cells (despite inducing brain-mediated inhibition of food intake), a feature that may worsen HFD-induced hyperinsulinemia and contribute to the observed differences in tumor immunogenicity in these 2 models. It would therefore be relevant to measure this serum parameter and at least discuss the putative contribution of hyperinsulinemia in T cell dysfunction (e.g.

Fisher et al. J Immunol 2017; Tsai et al Cell Metabolism 2018). Similarly, it is surprising that, by contrast to cholesterol, serum glucose levels, a key nutrient for both tumor and T cells, were not determined. Taken into account the central role of T cell-intrinsic metabolism in shaping their effector functions, one may question whether any cluster-specific differences in transcriptional ‘metabolic’ signature can be found in tumor-sorted CD8+ TIL from lean and obese mice. As such, a non-biased gene ontology analysis of the scRNAseq data would be useful, at least as supplementary data.

We agree that the role of circulating insulin and glucose in obese mice is an important variable to consider in relation to T cell function. To address this, we conducted new in vivo experiments and assessed fasting glucose and insulin levels in lean and obese mice. Results indicated no differences between cohorts, and these data are now provided (**Suppl. Fig. S1**) and are described in the revised Results section (line 218). Despite these similarities, we agree with the reviewer that differences in the nutrient composition between diets may still impact CD8+ TIL metabolism and effector function. As suggested by the reviewer, we have now performed non-biased gene ontology analysis of our scRNAseq data on total CD8+ TIL from lean and obese mice, indicating key differences in the regulation of several metabolic pathways (**Suppl. Fig. S2**). This analysis suggests that TIL in the obese tumor microenvironment fail to metabolically switch from oxidative phosphorylation to glycolysis, a process thought to be required for gain of effector function, providing new insight into a possible mechanism of obesity-related immune dysfunction, as discussed in the revised manuscript (line 235).

Minor

- Please provide the composition/reference of normal chow diet

We have provided the reference for the normal chow diet in the methods section (line 100).

- Line 228: “increased serum liver enzymes”, used as readout for HFD-induced liver damage

We have edited this passage in the Results section (line 218).

- Figure 3G: p-values not aligned

We have addressed this in Fig. 3G and also confirmed the accuracy of all other graphs.

- Figure 4D-E: color coding for lean OT-II mice may be confusing, with the same one used in panel A-C for WD-fed obese mice.

We agree with the reviewer’s suggestion and have changed the color of these points to green.

REVIEWERS' COMMENTS

Reviewer #1 (Remarks to the Author):

The revised manuscript contains highly impactful results regarding the effects of host obesity on the cancer immunoediting process and immune responses in the context of immune checkpoint blockade therapy. It also provides compelling data on the ability of healthy, diet-driven weight loss to restore CD8 T cell function and immunotherapeutic efficacy, relative to semaglutide-induced weight loss, which accomplishes neither. Additional data regarding the inability of atorvastatin to restore T cell function are ground-breaking and also clinically relevant. During the revision process, the authors worked hard to address each of our concerns experimentally and/or in the text, as necessary. We are pleased to recommend this manuscript be accepted for publication in Nature Communications. We predict it will have a substantial impact on the field of obesity-related cancer immunotherapy research.

Lyse A. Norian, Ph.D.

Henry Nnaemeka Ogbonna, M.S.

Reviewer #2 (Remarks to the Author):

The authors have thoroughly addressed all my concerns.

Reviewer #3 (Remarks to the Author):

Although the authors have circumnavigated a bit around the initial question on circulating insulin in semaglutide-treated mice and provided surprising data showing absence of hyperinsulinemia in their HFD-fed model, they have also addressed satisfactorily the other questions raised.

Response to Reviewer Comments:

Reviewer #1 (Remarks to the Author):

The revised manuscript contains highly impactful results regarding the effects of host obesity on the cancer immunoediting process and immune responses in the context of immune checkpoint blockade therapy. It also provides compelling data on the ability of healthy, diet-driven weight loss to restore CD8 T cell function and immunotherapeutic efficacy, relative to semaglutide-induced weight loss, which accomplishes neither. Additional data regarding the inability of atorvastatin to restore T cell function are ground-breaking and also clinically relevant. During the revision process, the authors worked hard to address each of our concerns experimentally and/or in the text, as necessary. We are pleased to recommend this manuscript be accepted for publication in Nature Communications. We predict it will have a substantial impact on the field of obesity-related cancer immunotherapy research.

Lyse A. Norian, Ph.D.

Henry Nnaemeka Ogbonna, M.S.

Response: We appreciate the reviewer's diligent work in helping to improve our manuscript and appreciate their enthusiasm for our findings.

Reviewer #2 (Remarks to the Author):

The authors have thoroughly addressed all my concerns.

Response: We are grateful for the reviewer's suggestions during the revision process and are glad that all concerns have been addressed.

Reviewer #3 (Remarks to the Author):

Although the authors have circumnavigated a bit around the initial question on circulating insulin in semaglutide-treated mice and provided surprising data showing absence of hyperinsulinemia in their HFD-fed model, they have also addressed satisfactorily the other questions raised.

Response: We thank the reviewer for their input on improving our manuscript and are pleased to have sufficiently addressed their questions.